# Evaluating the Impact of Statistical Bias Correction on Climate Change Signal and Extreme Indices in the Jemma Sub-Basin of Blue Nile Basin

Gebrekidan Worku Tefera [1,*], Yihun Taddele Dile [2] and Ram Lakhan Ray [1]

1   Cooperative Agricultural Research Center, College of Agriculture and Human Sciences, Prairie View A&M University, Prairie View, TX 77446, USA
2   College of Agriculture and Life Sciences, Texas A&M University, College Station, TX 77843, USA
*   Correspondence: gwtefera@pvamu.edu

**Abstract:** This study evaluates the effect of the statistical bias correction techniques of distribution mapping and linear scaling on climate change signals and extreme rainfall indices under different climate change scenarios in the Jemma sub-basin of the Upper Blue Nile Basin. The mean, cumulative distribution function (CDF), mean absolute error (MAE), probability of wet days ($Pr_{wet}$ (%)), and 90th percentile ($X_{90}$ (mm)) of observed rainfall and the regional climate model (RCM) simulations of rainfall with and without statistical bias correction were compared with the historical climate (1981–2005). For future (2071–2100) climate scenarios, the change in climate signal and extreme rainfall indices in the RCM simulations with and without bias correction were also evaluated using different statistical metrics. The result showed that the statistical bias correction techniques effectively adjusted the mean annual and monthly RCM simulations of rainfall to the observed rainfall. However, distribution mapping is effective and better than linear scaling for adjusting the probability of wet days and the 90th percentile of RCM simulations. In future climate scenarios, RCM simulations showed an increase in rainfall. However, the statistically bias-adjusted RCM outputs revealed a decrease in rainfall, which indicated that the statistical bias correction techniques triggered a change in climate signal. Statistical bias correction methods also result in changes in the extreme rainfall indices, such as frequency of wet days (R1mm), number of heavy precipitation days (R10mm), number of very heavy rainfall days (R20mm), and other intensity and frequency indices.

**Keywords:** statistical bias correction; climate change signal; extreme indices; regional climate models; Blue Nile Basin

## 1. Introduction

Climate models are essential to investigate the relative effect of human-induced emissions of greenhouse gases, such as carbon dioxide, methane, and nitrous oxide, on the climate system of the Earth [1,2]. Additionally, climate models are essential for studying the effects of climate change on the biophysical environment using different environmental models [3,4]. Over the past decade, various improvements have been observed in the science of climate modeling. For instance, the spatial resolution of climate models, particularly global climate models (GCMs), showed significant improvement from 1960 to the 2010s [1,5]. The commencement of Earth system models (ESMs), which integrate the biogeochemical cycle, aerosols, and anthropogenic sulfur dioxide emissions into the atmosphere–ocean GCMs, is another key proliferation in the evolution of climate modeling [6,7]). Despite several improvements, uncertainties remain in GCMs' simulations of the climate at the regional and basin scales, mainly due to coarse spatial resolutions and parameterization schemes [8,9]. As a result, downscaling GCM simulations and using regional climate model (RCM) simulations is recommended in order to obtain better climate information for regional- and local-scale climate impact studies [2,7].

Regional climate models have different added values compared to the GCMs. For instance, RCMs are better at simulating the climates of regional and local spatial scales, the climates of coastal and mountainous regions, and the parametrization of small-scale processes like convective precipitation [2,10,11]. Furthermore, RCMs generate detailed regional climate information and are superior to GCMs in capturing seasonal, short-duration, and extreme climate events [12]. However, the application of RCMs is still challenging since they use the GCMs' outer boundary conditions, and thus the driving GCM matters in the performance of the RCMs [13]. In the RCM simulations, systematic biases persist, and performance is inconsistent across regions and seasons [14–17]. This warrants using robust statistical bias correction techniques to adjust RCM simulations and study climate change in a particular region and/or season [18,19].

Statistical bias correction requires scaling climate model simulations to reproduce the observed climate [20–22]. Bias correction techniques adjust systematic biases of climate model simulations to reproduce observed climate values [23]. Bias correction also includes adjusting the systematic bias in the RCM model simulations using a transfer function to capture the statistical properties of observed climates, such as mean, variance, or distribution of observed values [14,20]. The fundamental idea of bias correction is developing a function that adjusts the climate model's simulation with observed counterparts [18,19]. Several bias correction methods, such as delta change [24] and linear scaling [25], only adjust the means of climate model simulations. Other methods, like distribution mapping, correct the means and frequencies of climate model simulation values [14,26,27].

The skill of different statistical bias correction techniques in correcting the mean, extremes, and frequency distribution of values of climate variables are different [23,27–29]. The distribution-based bias correction methods, such as distribution mapping and power transformation, are best at correcting frequency-based indices, and mean-based methods, such as linear and local intensity scaling, are best for mean and time-series-based indices in arid areas [28]. Similarly, the distribution-based method was superior to the mean-based bias correction method in correcting the magnitude of precipitation and wet-day frequencies over topographically structured terrain [30]. Most bias correction techniques show comparable performance in reproducing mean values [23,28,29].

On the other hand, bias correction methods may trigger biases [31–33]. For instance, the bias-corrected Weather Research and Forecasting (WRF) RCM resulted in a larger wet bias than the non-bias-corrected WRF simulation over Canada and Central North America [33]. In the southeastern USA, bias correction increased the difference between observed and simulated annual precipitation and overestimated the annual R-factor by an average of 137%. In contrast, non-bias-corrected data underestimated the R-factor by an average of 62% compared to the observed annual R-factor [32]. Statistical bias correction methods also have a limitation where there is high dependence on the quality of the observational data used to develop scale and shape parameters during bias correction [34]. Another limitation is that bias correction methods consider biases, and bias correction algorithms are stationary over time [20,32]. The stationarity and non-stationarity of the bias-adjusting algorithms are to be evaluated in different climate conditions using differential split-sample testing (DSST) and split-sample testing (SST) [35,36] before their use for climate change scenario development and climate change impact assessment. This is because the non-stationarity of the climate model simulations may reduce the effectiveness of bias correction methods, which further affects the output and thereby reduces weather and climate forecast and projection quality.

Bias correction techniques may also have limitations in terms of preserving the change signal, extremes, and quantiles of GCM and RCM simulations, thereby causing another chain of uncertainty in climate change scenario development [31,37,38]. In the Senegal River Basin, bias correction of raw RCM simulations causes a general dampening of the climate change signals and changes in heavy precipitation events [37]. The choice of statistical bias correction and downscaling techniques caused a change in climate signals, and even a reverse change signal, in the south-central region of the USA [38]. Similarly, it

was investigated that bias correction and spatial downscaling affect the heatwave frequency and reduce the temporal variability of most extreme indices in China [39]. On the other hand, a strong effect of observation data compared to the downscaling and bias correction techniques was investigated in different regions of the world [15,40].

Thus, it is essential to evaluate the effect of statistical bias correction techniques on climate change signals and the frequency of extreme values using different statistical metrics before using bias-corrected climate models' outputs for climate change impact assessment and adaptation-making systems. This promotes identifying statistical bias correction methods that preserve quantile changes and extremes. Even though the distribution-based method effectively captures the extreme distribution of different values, the quantile mapping method overestimates the precipitation extreme of the raw GCM simulations of the 2080s [14]. On the other hand, the detrended quantile mapping and quantile delta mapping techniques were effective and outperformed the standard quantile mapping technique in terms of preserving precipitation extremes projected by raw climate models in the Blue Nile River Basin [41]. Other studies have investigated the differences in statistical bias correction techniques regarding the preservation of quantiles and extremes of raw climate model outputs [23,42,43].

This study was conducted in the Jemma sub-basin of the Blue Nile Basin and focused on the following questions: (1) how are statistical bias correction techniques effective in adjusting the RCM simulations with observed data during the historical period? (2) what is the effect of statistical bias correction techniques on climate change signals and extreme rainfall events in the future period? Therefore, this study examined the bias-corrected and non-bias-corrected outputs of RCM simulations in the Jemma sub-basin in the historical climate scenario. We also compared the climate change signal and extreme rainfall values before and after bias correction in future climate scenarios, which were developed from multi-model simulations and emission scenarios of RCP4.5 and RCP8.5. This study is essential to determine the specific strengths and limitations of statistical bias correction techniques and to identify climate signals and extremes preserving statistical bias correction techniques. This study is also important for identifying possible uncertainties stemming from statistical bias correction during climate change projection, impact assessment, and adaptation decision analysis.

## 2. Materials and Methods

### 2.1. Study Area

The Jemma sub-basin is one of the sub-basins of the Upper Blue Nile Basin. This sub-basin has an area of about 15,000 km$^2$ and is located in the eastern part of the Upper Blue Nile Basin (Figure 1). The main river of the sub-basin is Jemma, while there are tributary rivers such as Beressa, Chacha, Robi-Gumero, and Moferwuha. The climate, particularly the rainfall of the sub-basin, is controlled by the transport of moisture from the equatorial East Pacific, the Indian Ocean, and the Arabian Peninsula [44]. The mean annual rainfall (1981 to 2014) ranges from 700 mm to 1500 mm, where the western and eastern parts of the sub-basin receive high rainfall while the northern part of the sub-basin receives low rainfall. From 1981–2014, the Jemma sub-basin showed an increasing trend in mean annual rainfall as well as extreme rainfall and temperature events [45]. The mean annual temperature is 9 °C in the northern and eastern parts and 24 °C in the western and central parts of the sub-basin.

The Jemma sub-basin contributes a significant amount of streamflow (14%) and sediment to the Upper Blue Nile River. The annual streamflow in 1996 and 1997 was 5844 million cubic meters and 2560 million cubic meters, respectively [46]. In this sub-basin, an annual sediment load of 21.2 million tons was measured from 1970–2010, which is the highest of all the sub-basins of the Upper Blue Nile Basin [47]. There are different and varied agroecological zones. The afro-alpine and sub-afro-alpine ecosystems are dominant in the eastern region. However, the central part of the sub-basin is characterized by cool sub-moist and temperate sub-moist highlands, while the western part of the sub-basin is

mainly under warm moist, temperate moist, and cool moist agroecologies. The elevation ranges from 1040–3840 m above sea level. The southern part is characterized by a plain with uniform topography and a gentle slope, while the central and eastern parts of the sub-basin are located at high elevation (≥3000 m.a.s.l) and have dissected terrains.

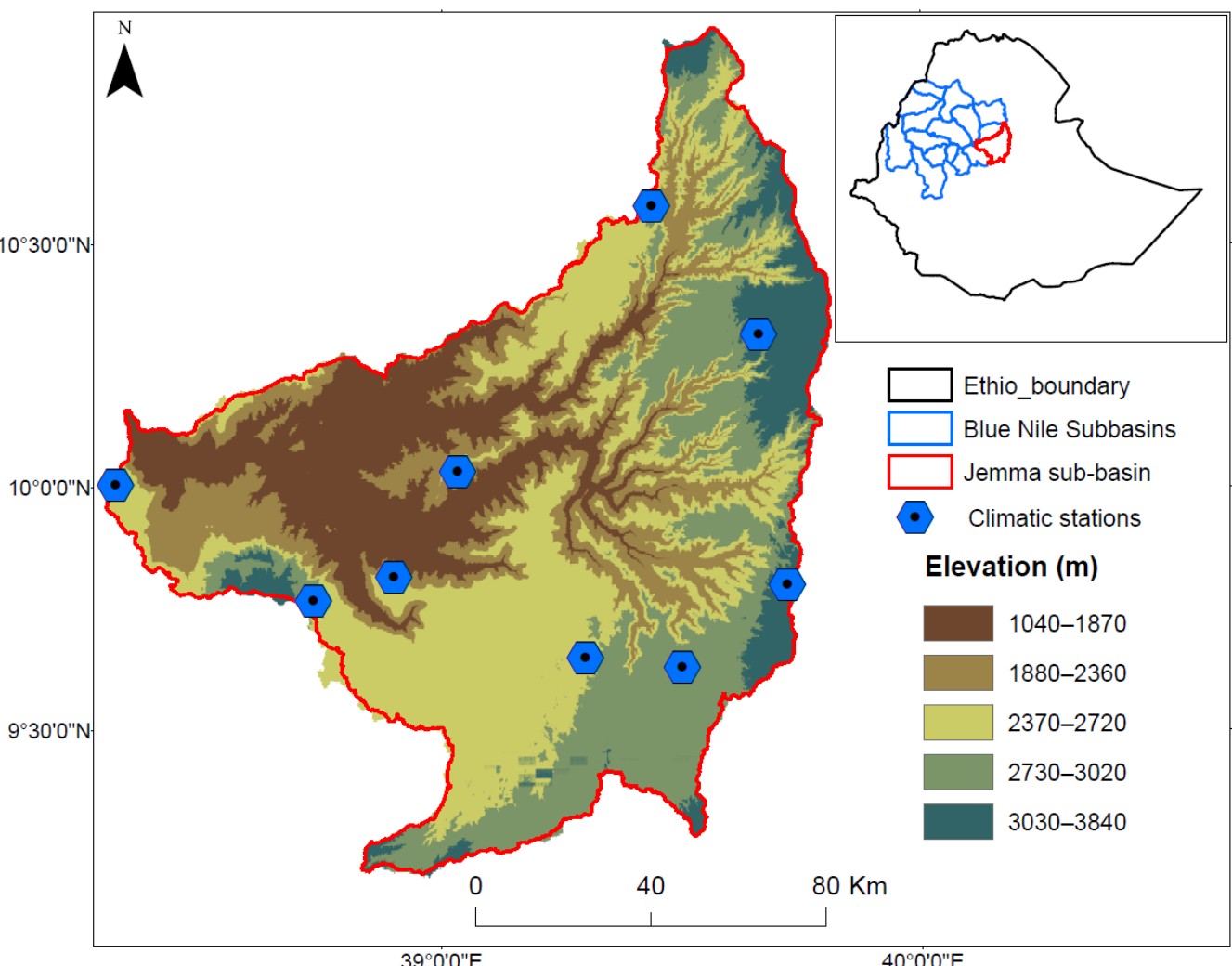

**Figure 1.** Location of Jemma sub-basin with reference to Ethiopia and the Upper Blue Nile Basin.

### 2.2. In Situ Climate Data

Historical (1981–2005) daily rainfall data was obtained from the National Meteorological Agency of Ethiopia. Nine out of thirteen climatic stations, with lower missing values (2–17%), were considered for this study. The climatic stations considered in this study represent the diverse topographic and agroecological zones that range from cold, moist sub-afro alpine to warm, sub-moist lowlands. In addition, the missing rainfall values of climatic stations were imputed using the multivariate imputation by chained equations (MICE) package, built into the R statistical software version R 3.2.3 [48]. The MICE package inputs the missing value of a single climate station by using the recorded values of other stations as predictors. This indicates that the MICE package creates multiple predictions to input each missing value. There was a need to prepare the data of all climatic stations, install the MICE and related packages in R, and input the missing values using the MICE package.

Quality control of the climate data, which includes managing errors and outliers, was carried out using the RClimDex [49]. Errors in the climate data, such as negative rainfall values and maximum temperatures less than minimum temperatures and outliers,

were corrected using the nearby stations. Besides outliers, values $\pm$ 4 times the standard deviation [50] were replaced by average values of the days before and after the outlier day.

### 2.3. Regional Climate Model Simulations Data

This study used historical and future RCM simulations of the Coordinated Regional Climate Downscaling Experiment (CORDEX) [51] driven by MPI-ESM-LR (ESM of the Max-Planck-Institut für Meteorologie ESM) GCM. The CCLM4 (COnsortium for Small-scale MOdeling (COSMO) Climate Limited-Area Model (CCLM) and REMO (Max Planck Institute Regional Model), which dynamically downscaled the MPI-ESM-LR, were considered in this study. These RCM simulations of CORDEX were selected because GCMs downscaled by CCLM4 and REMO were better at reproducing the mean and frequency of the Jemma sub-basin rainfall events than the Rossby Centre Regional Atmospheric Model fourth version (RCA4) [16]. The RCMs have spatial resolutions of $0.44° \times 0.44°$. Thus, historical (1981–2005) and future (2071–2100) raw simulations and the first ensemble member (r1) of these RCMs were used in this study.

### 2.4. Statistical Bias Correction Techniques

Linear scaling and distribution mapping statistical bias correction techniques were used to correct biases in historical and future (under RCP4.5 and RCP8.5) rainfall simulations of CCLM4 (MPI-ESM-LR) and REMO (MPI-ESM-LR). The linear scaling technique was selected because it is a mean-based statistical bias correction method. The distribution mapping method was selected because it adjusts the cumulative distribution function of climate model simulations toward the cumulative distribution function of observed values. This study considered the characteristics of statistical bias correction techniques, which are quite different and represent various mean-based and distribution-based bias correction techniques. For instance, the distribution mapping technique represents other methods, such as quantile–quantile mapping, probability mapping, and statistical downscaling [27]. Both statistical bias corection techniques determine a transfer function $h$ used to bias correct the RCM simulation $V_m$ such that the bias-corrected values equaled the observed values $V_o$ [18,19]. This function can be expressed as:

$$V_o = h(V_m) \tag{1}$$

The linear scaling technique bias corrects the biases of RCM simulations regarding rainfall using a multiplicative factor [25]. In linear scaling, the change factor that fits the RCM simulations to the observed counterparts is developed by comparing the observed rainfall with the corresponding historical RCM simulations. This statistical bias correction technique adjusts biases in the mean, but has limitations in terms of correcting biases in the frequency and intensity of rainfall.

The distribution mapping method adjusts the mean and cumulative distribution function (CDF) of RCM rainfall simulations toward the mean and CDF of observed rainfall using a transfer function that fits the occurrence of different rainfall values. The frequency of different rainfall values, variance, standard deviation, and extreme values of RCM simulations are corrected through the distribution mapping technique [20,27,52]. This bias correction technique applies Gamma distribution to adjust the CDF of rainfall of RCM simulations with the CDF of observational rainfall [19]. Distribution mapping uses an RCM-specific rainfall threshold [27] to adjust the frequencies of rainfall days. In this technique, the function which adjusts the RCM simulations of rainfall based on the observed rainfall is given as:

$$V_o = F_o - 1(F_m(V_m)) \tag{2}$$

where

$V_o$ is an observed variable;
$V_m$ is modeled variable;
$F_m$ is the CDF related to $V_m$;

$F_o{}^{-1}$ is the inverse CDF of $V_o$ [19].

### 2.5. Future Rainfall and Temperature Extremes Analysis

The Expert Team on Climate Change Detection and Indices (ETCCDI) has developed 27 rainfall and extreme temperature indices, which measure the intensity, frequency, and duration of rainfall and temperature extremes [50]. This study used the ETCCDI indices to measure the trend of rainfall extremes in historical and future climate scenarios. The indices included in this study measure the intensity, frequency, and duration of rainfall extremes. Daily observed (1981–2005), raw, and bias-corrected RCM simulations of historical (1981–2005) and future (2071–2100) rainfall were used to calculate the extreme rainfall indices. The R built-in interface, i.e., the RClimDex 1.1 [49], was used to detect indices of rainfall and temperature extremes. Eventually, we analyzed the change in extreme rainfall indices with and without bias correction in the historical and future RCM simulations.

### 2.6. Evaluation Techniques

The cumulative distribution function (CDF) of the observed RCM simulations and the bias-corrected RCM output were compared for the historical period. Besides the 90th percentile ($X_{90}$ (mm)) and wet-day probability ($Pr_{wet}$ (%)) of raw and bias-corrected RCMs, rainfall was compared against the observed historical rainfall counterparts. The box plots were used to analyze the median, maximum, minimum, 25th, and 75th percentiles and outliers in the observed, raw, and bias-adjusted RCM rainfall simulations. The mean absolute error (MAE) was also used to analyze the RCM rainfall simulation before and after statistical adjustment at the monthly scale; this comprised the wet and dry seasons. The spatial pattern of observed, raw, and bias-adjusted RCM outputs were compared to analyze the effect of bias-adjusting methods on the spatial variation of rainfall in the sub-basin.

In both the baseline and future climate change scenarios, we compared the change signals, frequency, and intensity of extreme rainfall events in the raw and bias-corrected RCM simulations. The change signal, which is the difference between the rainfall of future raw and bias-corrected RCMs output (2071–2100), was computed from the observed historical rainfall (1981–2005). Similarly, the change in extreme rainfall events was computed using the extreme values of future raw and bias-corrected RCM output and the observed historical extreme rainfall values.

## 3. Results and Discussion

### 3.1. Evaluation of Bias Correction Techniques

Figures 2 and 3 show that distribution mapping and linear scaling bias correction techniques can effectively adjust the mean annual and monthly rainfall of raw RCM simulations to the observed rainfall. The CCLM4 (MPI-ESM-LR) and REMO (MPI-ESM-LR) models overestimated the mean annual rainfall by 64 mm and 91 mm, respectively. Furthermore, the median, 25th, and 75th percentiles, as well as the maximum mean annual rainfall in the raw RCM simulations, were higher than the observed rainfall. The performance of RCM simulations was also found to be elevation-dependent, where there is underestimation and overestimation following the variation in elevation. The CCLM4 (MPI-ESM-LR) overestimated and underestimated the mean annual rainfall of higher-elevation (northern and eastern part of the sub-basin (Figure 1)) and lower-elevation (western) areas of the Jemma sub-basin, respectively. On the other hand, the REMO (MPI-ESM-LR) underestimated the annual rainfall of the eastern part of the sub-basin and overestimated the mean annual rainfall of the central part of the Jemma sub-basin. Such elevation-dependent biases of RCM simulations may be attributed to the driving GCM being used as a boundary condition or the RCM parametrizations of clouds in the higher- and lower-elevation areas. Concurrently, the limitations of RCMs in simulating drizzle rainfall events (<1 mm) and heavy rainfall events (>10 mm), as well as biases mainly regarding variations in elevation, were investigated in the Upper Blue Nile Basin [53] and the Ethiopian highlands [54]. Thus, statistical pre-processing techniques table to adjust elevation-dependent under- and

overestimation of precipitation by the RCM simulations should be considered before using RCM simulations for further applications.

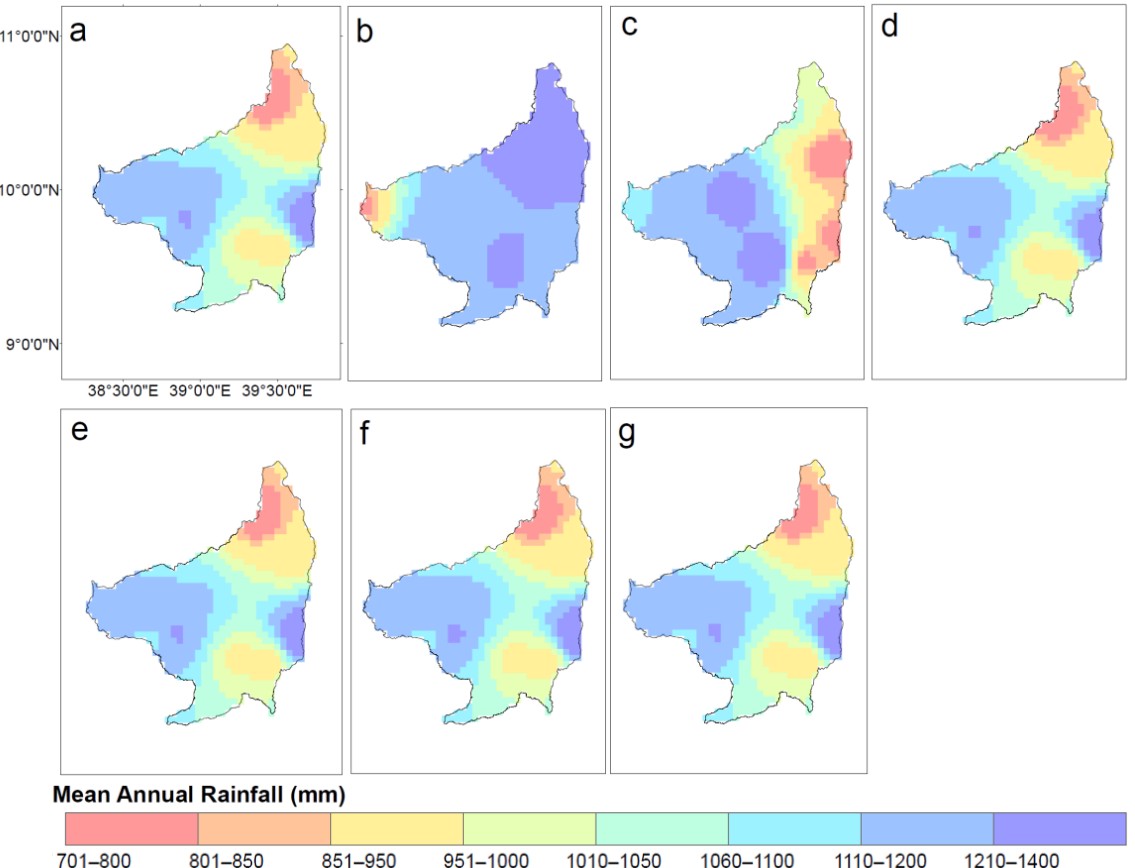

**Figure 2.** Mean annual rainfall of observed and RCM simulations before and after statistical bias correction for the historical period (1981–2005). (**a**) Observed, (**b**) CCLM4 (MPI-ESM-LR), (**c**) REMO (MPI-ESM-LR), (**d**) CCLM4 (MPI-ESM-LR)-DM, (**e**) CCLM4 (MPI-ESM-LR)-LS, (**f**) REMO (MPI-ESM-LR)-DM, (**g**) REMO (MPI-ESM-LR)-LS. The DM and LS represent distribution mapping and linear scaling bias correction methods, respectively.

The spatial distribution of mean annual rainfall of RCM simulations was adjusted, and the observed mean annual rainfall was reproduced. The overestimation and underestimation of rainfall by the RCM simulations in different areas of the sub-basin were corrected and the observed rainfall was adequately captured. The statistical bias correction techniques were efficient and revealed comparable skills in preserving the spatial distribution of mean annual rainfall of the sub-basin (Figure 2). The boxplots show that the median values were improved and able to reproduce the observed median rainfall after the RCM simulations were bias-adjusted. The statistical bias correction methods trigger a change in the minimum, maximum, 25th, and 75th percentiles of the mean annual rainfall (Supplementary Figure S1). For instance, the higher 25th percentile in the raw RCM simulations was reduced, and fell even lower than its observed counterparts after bias correction. This indicates that statistical bias correction techniques effectively correct mean values and capture the spatial distribution of the observed rainfall. On the other hand, this also implies that statistical bias correction may strongly change the raw RCM simulations and trigger a change in the frequency of high- and low-rainfall events.

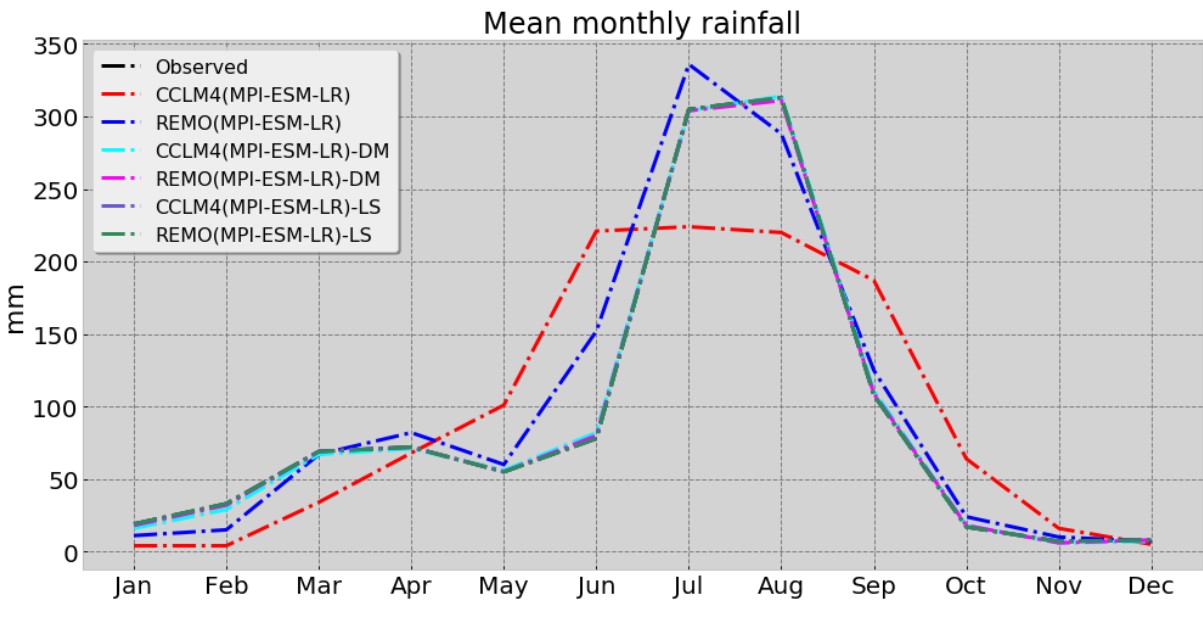

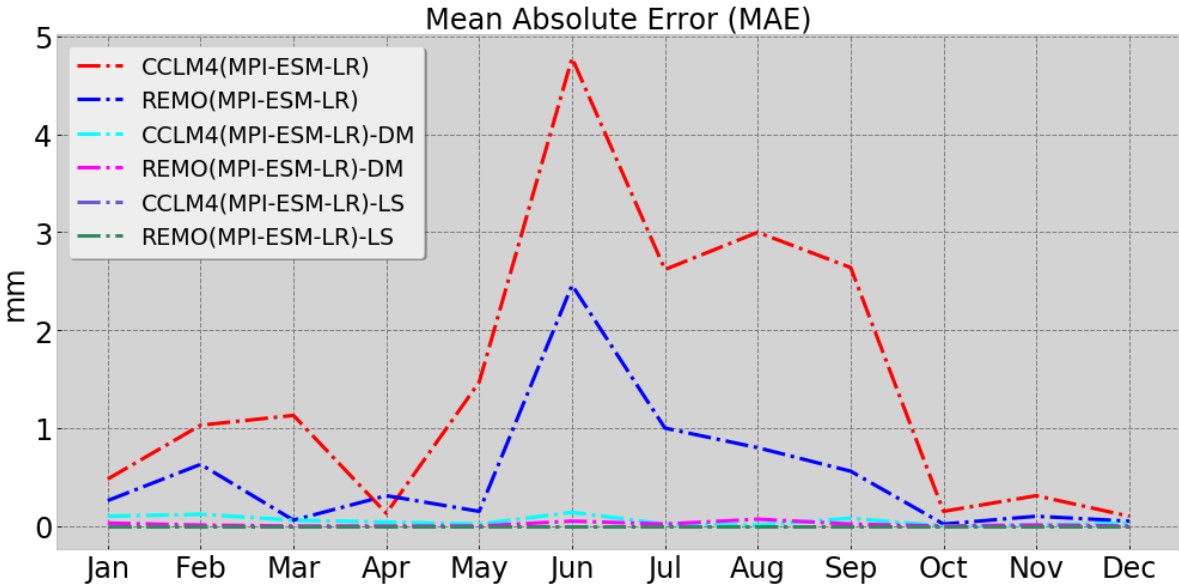

**Figure 3.** Mean monthly rainfall and mean absolute error (MAE) of RCM simulations before and after statistical bias correction for the historical period (1981–2005). The DM and LS represent distribution mapping and linear scaling bias correction methods, respectively.

The mean monthly rainfall of RCM simulations in different months reveals that linear scaling and distribution mapping techniques are effective and show comparable performance in adjusting the biases of the RCM simulations (Figure 3). The raw RCM simulations were characterized by overestimating and underestimating observed rainfall in different months. The CCLM4 (MPI-ESM-LR) simulated higher rainfall in the April–June and September–November months. However, this RCM underestimated the main rainy season's (July–August) rainfall. The REMO (MPI-ESM-LR) performed better in simulating the monthly rainfall. However, REMO (MPI-ESM-LR) also overestimated the May–June rainfall and underestimated the July–September rainfall. Both RCM simulations had limitations in terms of reproducing the rainfall of the main rainfall season (June–September). This corroborates that the RCM simulations have limitations in adequately simulating the seasonal variation of rainfall. Like elevation-dependent biases (Figure 2), the limitations of

RCM simulations in simulating the seasonal variation of rainfall may be attributed to the RCMs' failure to parametrize local topography or GCMs with lower spatial resolution.

Concurrently with the mean annual and monthly simulations, the raw RCM are characterized by higher MAE (up to 4.78 mm/day) in some months. In the RCM simulations, higher MAE (0.56 to 4.78) is detected during the main rainfall seasons (June to September) than during dry seasons. The RCM simulations after statistical bias correction showed lower MAE in the wet and dry months of the sub-basin (Figure 3). The outputs after bias correction had MAE values of less than 0.14 mm/day for all months. Particularly, the linear scaling method had high skill in MAE (MAE $\leq$ 0.001) compared to the distribution mapping method. This indicates the significant contribution of statistical bias correction techniques to reducing uncertainties related to monthly rainfall simulation. The difference in MAE between raw and bias-adjusted RCM outputs was higher and statistically significant ($\leq$0.01) than the difference between RCM types.

Unlike the mean-based metrics, the statistical bias correction methods showed differences of skill in distribution and frequency-based metrics such as CDF, wet-day probability (a), and 90th percentile ($X_{90}$). The CDF of the observed rainfall and RCM simulations (before and after bias correction) corroborated the added value of statistical bias correction techniques for adjusting the distribution of RCM simulations to the observed counterparts (Figure 4a,b). The bias-adjusted RCM outputs captured the CDF of observed rainfall more effectively than the raw RCM simulations. The RCM type also influenced CDF fitting through bias correction methods. For instance, the linear scaling method reduced the proportion of high-rainfall events from 19% to 13% in the CCLM4 (MPI-ESM-LR) model (Figure 4b). The CDF of REMO (MPI-ESM-LR) simulation was better at reproducing the CDF of observed rainfall than the CCLM4(MPI-ESM-LR) model. However, the RCM simulations both underestimated and overestimated the CDF of the observed rainfall. On the other hand, the REMO (MPI-ESM-LR) RCM bias, adjusted using the distribution mapping method, showed a superior performance in capturing the CDF of the observed rainfall. According to the raw and distribution mapping output of REMO (MPI-ESM-LR), the proportion of high rainfall ($\geq$10 mm/day) was about 15% of the total rainfall (Figure 4b). For the observed, raw, and bias-corrected outputs, the proportion of dry days and drizzle rainfall events was 42%, while the proportion of high rainfall ($\geq$10 mm/day) was 20% (Figure 4a,b).

The skill of bias correction techniques in correcting the wet-day probability and 90th percentile indicates the effect of these bias correction methods on the distribution and extreme values of RCM simulations. The wet-day probability and 90th percentile in Figure 5a,b reveal that statistical bias-adjusting methods added value in reducing the biases of RCMs when simulating the wet-day probability and 90th percentile. The RCMs (REMO (MPI-ESM-LR) and CCLM4 (MPI-ESM-LR)) overestimated the probability of wet days (Figure 5a). The linear scaling method struggled to adjust the probability of wet days to the observed counterparts. A high number of wet days were simulated in the raw and bias-corrected outputs by linear scaling methods. This result specifies that mean-based bias correction methods, like linear scaling, have limitations in terms of correcting low-rainfall (drizzle) days. In contrast, the distribution mapping method effectively reproduced the wet-day probability of observed rainfall from the RCM simulations. The RCM simulations underestimated the 90th percentile of rainfall in the wet seasons of the sub-basin. The distribution mapping method fit the 90th percentile of RCM simulations better than the observed rainfall counterpart. The linear scaling method also improved the 90th percentile simulation of the RCMs. However, this statistical post-processing technique showed a lesser ability to fit the 90th percentile compared to the distribution mapping method.

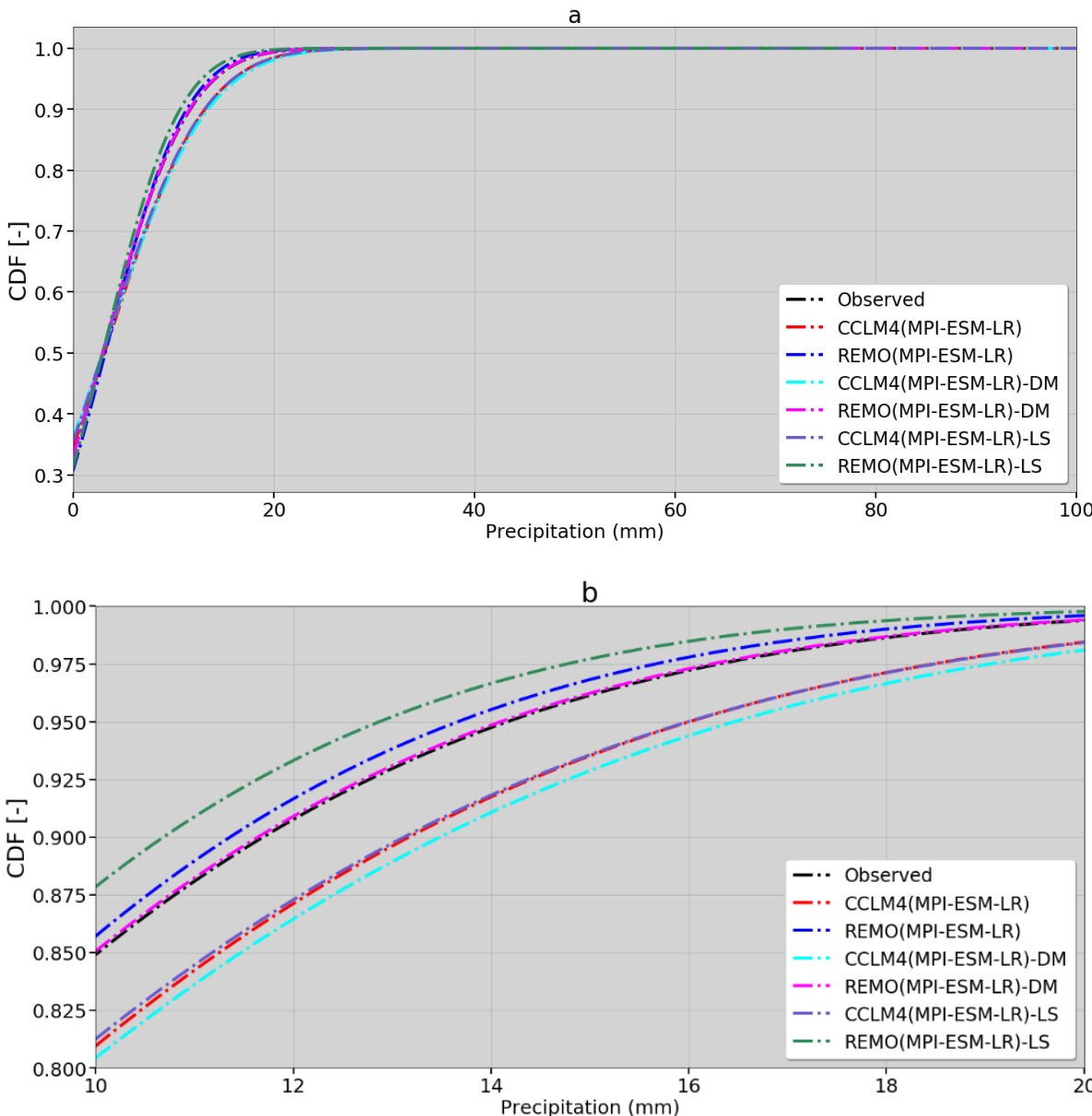

**Figure 4.** Cumulative distribution function (CDF) of observed and RCM simulations before and after statistical bias correction in the historical period (1981–2005): (**a**) presents the CDF of rainfall amount from 0 to 100 mm/day, while (**b**) shows the CDF of heavy rainfall ($\geq$10–20 mm/day). The DM and LS represent the distribution mapping and linear scaling bias correction methods, respectively.

The performance of statistical bias correction techniques in mean-based metrics, such as in capturing the mean annual and monthly values and MAE and distribution-based metrics, i.e., wet-day probability, the 90th percentile, and the CDF of observed rainfall, is comparable with other studies. For instance, the added value of statistical bias correction techniques was investigated by past studies [14,19,27,28]. Analogous to this study, there are studies which found underestimation and overestimation of observed values due to statistical bias correction [31–33,37]. For instance, there was a 19.2% difference between bias-corrected climate model simulation and observed annual precipitation, while the difference between raw climate model simulation and observed annual precipitation was 3.5% [32]. Such differences in the skill of bias correction techniques could be partly attributed to the quality of the observed data.

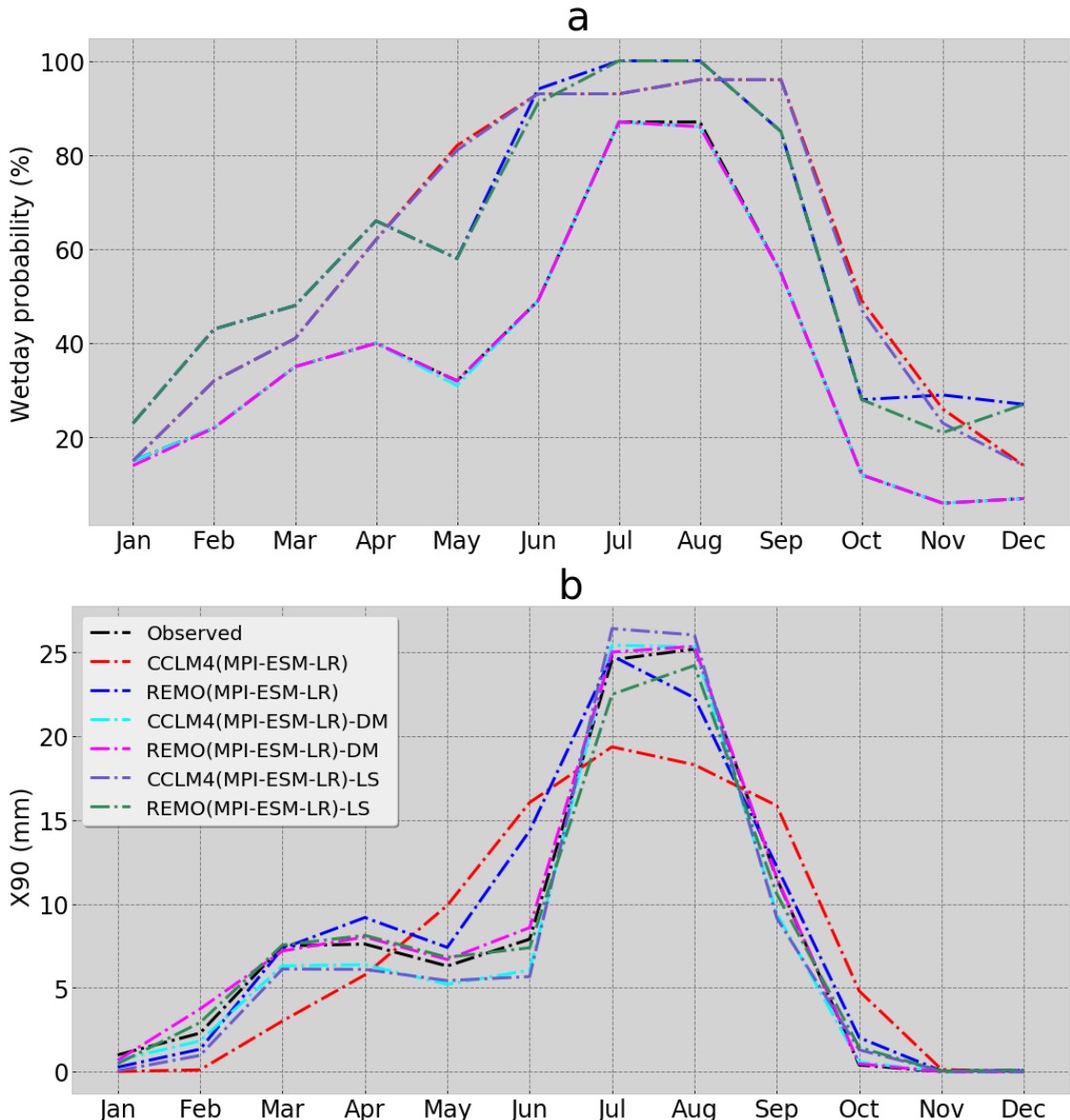

**Figure 5.** Wet-day probability (**a**) and 90th percentile ($X_{90}$) (**b**) of observed and RCM simulations before and after statistical bias correction in the historical period (1981–2005). The DM and LS represent the distribution mapping and linear scaling bias correction methods, respectively.

### 3.2. Effect of Bias Correction Techniques on Climate Change Signal

The RCM simulations of the future climate before and after bias correction were compared to evaluate the effect of bias correction on climate change signals and extreme indices. The raw RCM simulations consistently projected an increase in mean annual rainfall for the future climate under both RCP4.5 and RCP8.5 emission scenarios (Figure 6). With no difference in the RCPs, both CCLM4 (MPI-ESM-LR) and REMO (MPI-ESM-LR) simulated an increase in the mean annual rainfall on the order of 3% to 35% (Table 1). The REMO (MPI-ESM-LR) model projected higher rainfall for the future climate than the observed and CCLM4 (MPI-ESM-LR) rainfall. However, the bias correction methods triggered a reduction in future mean annual rainfall, but at different magnitudes. After applying the distribution mapping technique, the mean annual rainfall of 1034 mm and 1240 mm, according to CCLM4 (MPI-ESM-LR) and REMO (MPI-ESM-LR), were reduced to 829 mm and 1166 mm, respectively.

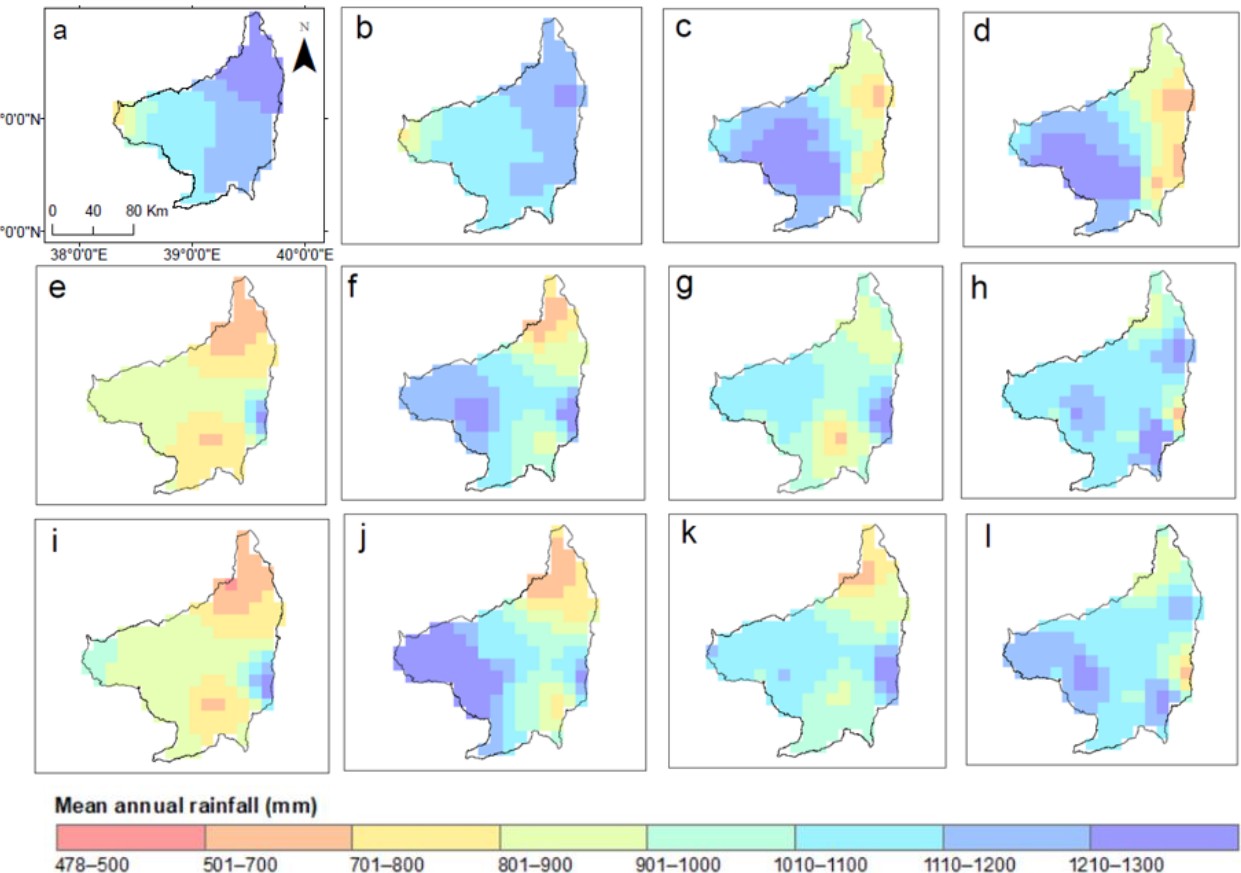

**Figure 6.** Mean annual of RCMs before and after statistical bias correction in 2071–2099. (**a**) CCLM4(MPI-ESM-LR)RCP4.5, (**b**) CCLM4(MPI-ESM-LR)RCP8.5, (**c**) REMO(MPI-ESM-LR)-RCP4.5, (**d**) REMO(MPI-ESM-LR)-RCP8.5, (**e**) CCLM4(MPI-ESM-LR)-RCP4.5-DM, (**f**) REMO(MPI-ESM-LR)-RCP4.5-DM, (**g**) CCLM4(MPI-ESM-LR)-RCP4.5-LS, (**h**) REMO(MPI-ESM-LR)-RCP4.5-LS, (**i**) CCLM4(MPI-ESM-LR)-RCP8.5-DM, (**j**) REMO(MPI-ESM-LR)-RCP8.5-DM, (**k**) CCLM4(MPI-ESM-LR)-RCP8.5-LS, and (**l**) REMO(MPI-ESM-LR)-RCP8.5-LS. The DM and LS represent the distribution mapping and linear scaling bias correction methods, respectively.

**Table 1.** Future (2071–2100) mean annual rainfall (mm) of RCMs before and after statistical bias correction. The bracket is the change signal (%) in mean annual rainfall between the future (2071–2100) and observed rainfall (1981–2005). DM and LS represent the distribution mapping and linear scaling bias correction methods, respectively.

| Emission Scenario | RCMs | Mean Annual Rainfall (mm) |
|---|---|---|
| RCP4.5 | Observed | 1001 (-) |
| | CCLM4(MPI-ESM-LR) | 1034 (3) |
| | REMO(MPI-ESM-LR) | 1240 (24) |
| | CCLM4(MPI-ESM-LR)-DM | 829 (−17) |
| | REMO(MPI-ESM-LR)-DM | 1166 (16) |
| | CCLM4(MPI-ESM-LR)-LS | 875 (−13) |
| | REMO(MPI-ESM-LR)-LS | 967 (−3) |

**Table 1.** *Cont.*

| Emission Scenario | RCMs | Mean Annual Rainfall (mm) |
|---|---|---|
| | CCLM4(MPI-ESM-LR) | 1035 (3) |
| | REMO(MPI-ESM-LR) | 1348 (35) |
| RCP8.5 | CCLM4(MPI-ESM-LR)-DM | 858 (−14) |
| | REMO(MPI-ESM-LR)-DM | 1344 (34) |
| | CCLM4(MPI-ESM-LR)-LS | 879 (−12) |
| | REMO(MPI-ESM-LR)-LS | 1078 (8) |

Statistical bias correction techniques resulted in changes in climate signals, and even reverse climate signals, in some RCM simulations (Table 1 and Figure 7). For instance, the distribution mapping bias correction method changed the positive rainfall signal in CCLM4 (MPI-ESM-LR) to a negative rainfall change signal under the RCP4.5 and RCP8.5 emission scenarios. Better than linear scaling, the distribution mapping at least preserved the sign of the climate change signal in REMO (MPI-ESM-LR) under the RCP4.5 and RCP8.5 emission scenarios. The linear scaling technique triggered a change in the positive climate change signal in the CCLM4 (MPI-ESM-LR) and REMO (MPI-ESM-LR) RCMs into a negative rainfall change signal under RCP4.5. The linear scaling and distribution mapping techniques revealed a comparable effect on climate change signals under RCP8.5. However, the distribution mapping method was found to be slightly better at preserving the climate change signal of the RCM simulations than the linear scaling method. In most cases, the positive rainfall signal in the REMO (MPI-ESM-LR) model was not changed into a negative rainfall signal (Table 1 and Figure 7). This reveals that the climate change signal is sensitive not only to the choice of bias correction techniques, but also to the type of RCM simulations.

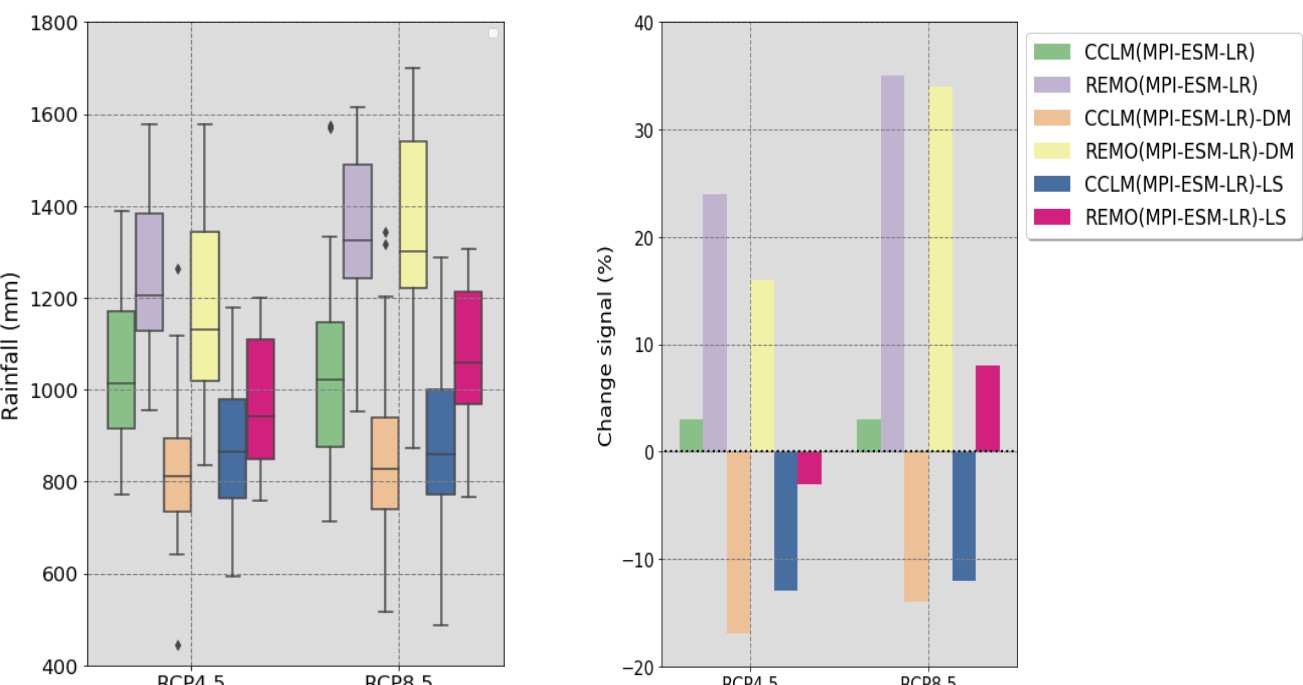

**Figure 7.** Boxplots and climate change signals of the annual rainfall of RCM simulations with and without statistical bias correction for 2071–2099. The DM and LS represent the distribution mapping and linear scaling bias correction methods, respectively. In the boxplots, the whiskers indicate the minimum and maximum of rainfall; the horizontal lines represent the 25th percentile, the median, and the 75th percentile from the bottom to the top of each box plot, and the diamond sign indicates outliers.

The result of this study corroborates that statistical bias correction is not the only source of uncertainty in climate change projection. RCM and driving GCM are other sources of variation. Similarly, other studies have explored a change in the climate change signal after applying statistical bias correction. For instance, the statistical bias correction of RCM simulations triggers a general dampening of the climate change signals in the Senegal River Basin [37], and a modification of climate change signal owing to bias correction was investigated in Swiss climate stations [55]. Other studies [14,15,41] identified a difference among the statistical bias-adjusting techniques in terms of preserving climate change signals.

### 3.3. Effect of Bias Correction Techniques on Rainfall Extreme Indices

In the historical climate, statistical bias correction techniques had a divergent effect on the frequency of wet days (R1mm), the number of heavy precipitation days (R10mm), and the number of very heavy precipitation days (R20mm) (Figure 8). The RCM simulations showed a high frequency of R1mm compared to the observed and bias-corrected RCM outputs. The distribution mapping method corrected the overestimation of R1mm in the RCM simulations. The R1mm days in the RCMs, bias-adjusted by the distribution mapping method, showed better agreement with the observed R1mm days (Figure 8). There were 111 and 109 R1mm days/year in the observed and CCLM4 (MPI-ESM) models, bias-adjusted by the distribution mapping method, respectively. This indicates that the scaling parameters used for the distribution mapping method were effective in fitting low-intensity rainfall values of RCM simulation. However, the linear scaling showed a higher number of R1mm days than were observed, and there were comparable numbers of R1mm days in the linear scaling outputs and RCM simulations. There were 121 and 123 R1mm days in the CCLM4 (MPI-ESM-LR) and REMO (MPI-ESM-LR) models, bias-corrected by the linear scaling method. Both statistical bias correction techniques added value in terms of adjusting the frequency of R1mm. However, the distribution mapping method was superior for adjusting R1mm days. This indicates that the linear scaling and distribution mapping techniques apply different factors to remove drizzle rainfall values. It is also noteworthy to remember that only the distribution mapping method used an RCM-specific precipitation threshold to adjust the frequency of wet days.

The CCLM4 (MPI-ESM) and REMO (MPI-ESM-LR) models showed a lower and higher number of R10mm days than the observed rainfall, respectively. In the historical climate, there were 38, 34, and 44 R10mm days/year in the observed, CCLM4 (MPI-ESM), and REMO (MPI-ESM-LR) models. Bias correction techniques triggered a reduction in R10mm days from the raw RCM simulations. Both distribution mapping and linear scaling techniques reduced the R10mm days identified by the RCM simulations. The only exception was distribution mapping which increased the minimum, maximum, and percentiles of the R20mm days compared to the RCM simulations. Unlike R10mm, the distribution mapping method increased the number of R20mm days compared to the RCM simulations. There were 15, 13, 13, 14, 14, 12, and 11 R20mm days/year in the observed, CCLM4 (MPI-ESM-LR), REMO (MPI-ESM-LR), CCLM4 (MPI-ESM-LR)-DM, REMO (MPI-ESM-LR)-DM, CCLM4 (MPI-ESM-LR)-LS, and REMO (MPI-ESM-LR)-LS simulations, respectively. This also indicates that the type of RCM was another factor determining the number of extreme rainfall values before and after bias adjustment. However, the significant difference ($\leq 0.05$) in the number of R20mm days was due to the choice of a bias correction method over the RCM types, where the distribution mapping technique was characterized by a high frequency of very heavy rainfall days [14], also investigated the quantile delta mapping technique and how it has estimated the frequency of extreme precipitation events to be far higher than the raw GCM simulation.

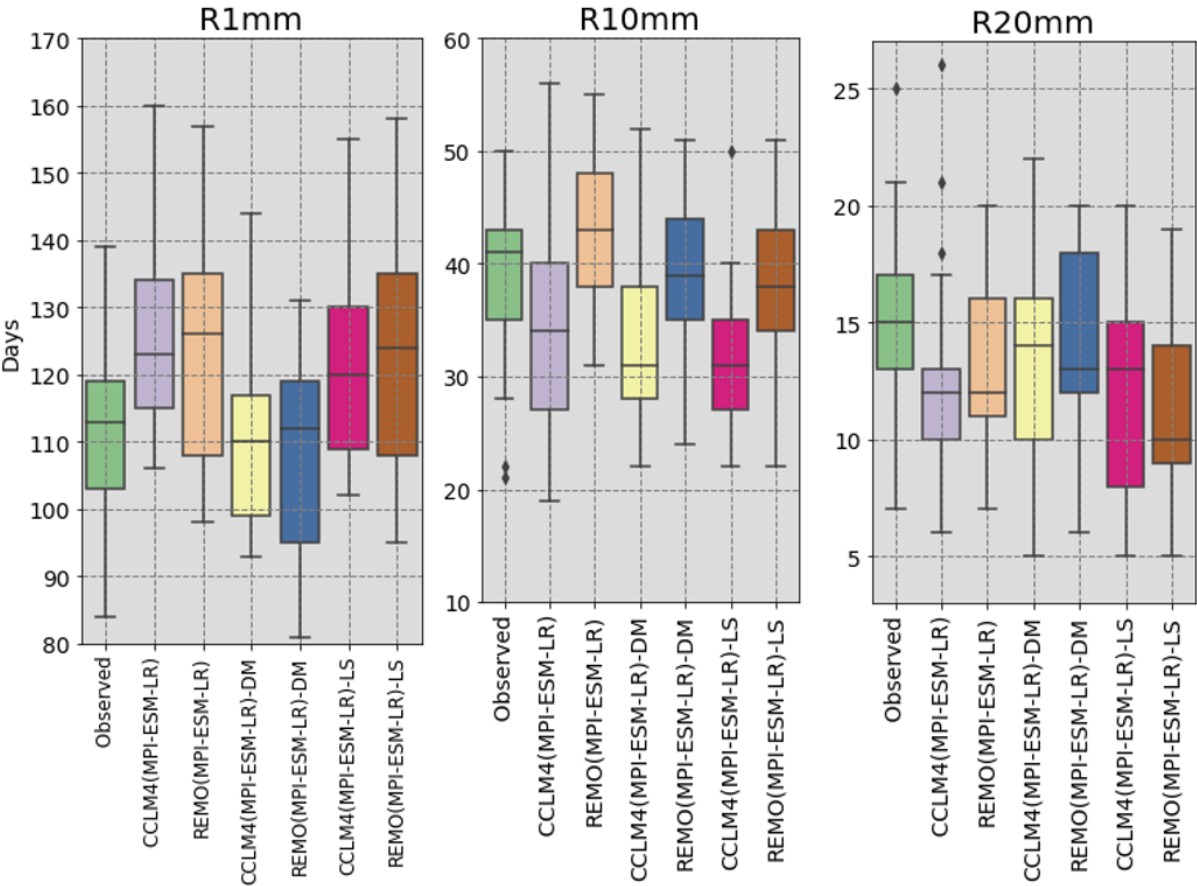

**Figure 8.** Boxplots of rainfall extremes of observed and RCM simulations, with and without bias correction, during the historical period (1981–2005). The DM and LS represent the distribution mapping and linear scaling bias correction methods, respectively. In the boxplots, the whiskers indicate the minimum and maximum of rainfall; the horizontal lines represent the 25th percentile, the median, and the 75th percentile from the bottom to the top of each box plot, and the diamond sign indicates outliers.

In the future climate change scenarios, the statistical bias correction methods triggered a change in the rainfall extremes of the RCM simulations (Figure 9). The distribution mapping method significantly reduced R1mm days compared to the RCM simulations under both the RCP4.5 and RCP8.5 emission scenarios. There were 117 and 115 R1mm days/year in the CCLM4 (MPI-ESM-LR) and REMO (MPI-ESM-LR) simulations under RCP4.5, respectively. However, the frequency of R1mm was reduced to 63 and 92 days/year in the CCLM4 (MPI-ESM-LR) and REMO (MPI-ESM-LR) simulations bias-corrected by the distribution mapping method and under RCP4.5. The linear scaling method preserved the RCM simulations of wet days.

Both bias correction techniques resulted in a dampening of R10mm days in the future period. However, the distribution mapping technique better preserved the R10mm days simulated by the RCMs. The frequency of R10mm was 30 and 46 days/year according to the CCLM4 (MPI-ESM-LR) and REMO (MPI-ESM-LR) simulations under RCP4.5, respectively. On the other hand, under a similar emission scenario (RCP4.5), the frequency of R10mm was 27 days and 41 days/year according to the CCLM4 (MPI-ESM-LR) and REMO (MPI-ESM-LR) simulations bias-corrected by the distribution mapping method. The linear scaling technique resulted in a higher reduction in R10mm days. The frequency of R10mm days decreased to 24 and 33 days/year according to the CCLM4 (MPI-ESM-LR) and REMO (MPI-ESM-LR) simulations bias-corrected by the linear scaling method and under RCP4.5. The distribution mapping technique also better preserved the R20mm days simulated by

the RCMs than the linear scaling. Like R10mm, linear scaling triggered a higher reduction in R20mm days. In general, the impact of bias correction on R10mm and R20mm days was more pronounced than the difference in RCM types and emission scenarios. A change in the frequency of projected extreme events has also been identified in other studies [14,37,38]. This indicates to what degree bias correction techniques modify extreme events in climate change projection, which further post-impact assessment and adaptation decisions, such as designing hydraulic structures to buffer the impact of climate extremes.

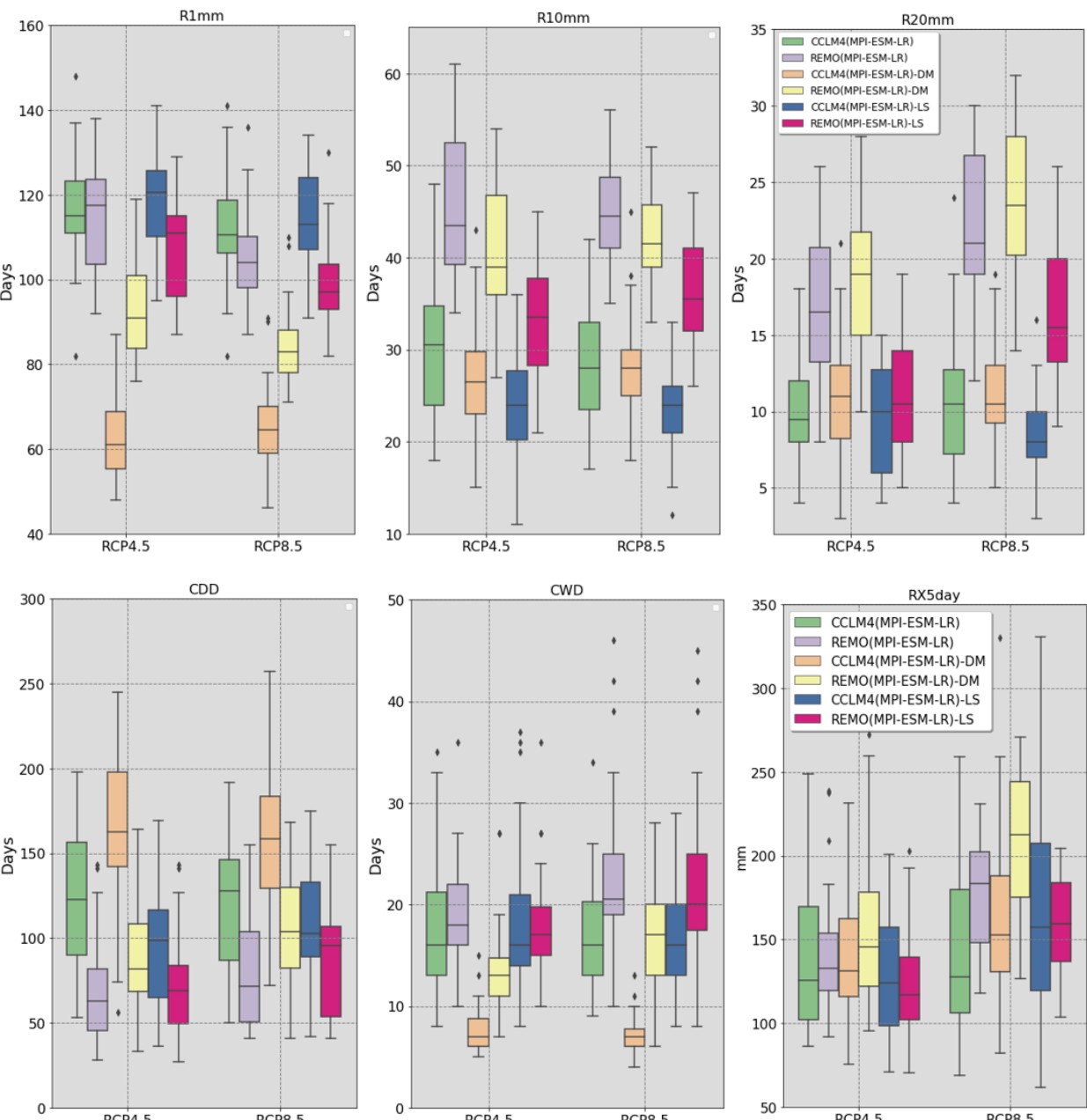

**Figure 9.** Boxplots of rainfall extremes of RCM simulations, with and without statistical bias correction, for the future period (2071–2100). The DM and LS represent the distribution mapping and linear scaling bias correction methods, respectively. In the boxplots, the whiskers indicate the minimum and maximum of rainfall; the horizontal lines represent the 25th percentile, the median, and the 75th percentile from the bottom to the top of each box plot, and the diamond sign indicates outliers.

The bias correction techniques also caused a change in the frequency of the consecutive dry days (CDD) and consecutive wet days (CWD) simulations by the RCMs. The

distribution mapping technique increased the CDD of the RCM simulations without excluding the RCMs and emission scenarios. However, the linear scaling reduced the CDD simulated by CCLM4 (MPI-ESM-LR) under the RCP4.5 and RCP8.5 emission scenarios. There were 162 and 88 CDDs/year in the CCLM4 (MPI-ESM-LR) and REMO (MPI-ESM-LR) simulations bias-corrected by the distribution mapping method under RCP4.5 emission scenarios. In contrast, there were 94 and 73 CDDs/year in the CCLM4 (MPI-ESM-LR) and REMO (MPI-ESM-LR) simulations bias-corrected by the linear scaling technique and under RCP4.5 emission scenarios. Conversely, the distribution mapping triggered a significant reduction in CWD of RCM simulations under both emission scenarios. This indicates that the distribution mapping method applies scale parameters that remove low-intensity rainfall days, which further results in a reduction in CWD and an increase in the frequency of CDD. The linear scaling technique preserves the CWD of RCM simulations under both emission scenarios.

## 4. Conclusions

This study evaluates the effect of mean-based and distribution-based statistical bias correction techniques on climate change signals as well as the frequency and intensity of extreme rainfall events in the historical and future periods in the Jemma sub-basin of the upper Blue Nile Basin. The RCMs are characterized by overestimation and underestimation in different sub-basin areas. Furthermore, a difference in terms of simulating rainfall at different elevations of the sub-basin among the RCMs was identified. The RCMs also struggle to reproduce the seasonal variation of rainfall of the sub-basin. This could be associated with uncertainties derived from the initial boundary condition (GCMs) or the difference among the RCMs in parameterizing convective clouds at different elevation classes of the studied sub-basin. Convective-permitting modeling schemes may capture such biases by effectively characterizing clouds and convective processes in high-elevation areas. In addition, further improvements to the spatial resolution of regional climate models may reduce elevation-dependent biases and simulate the seasonal variation of rainfall.

It has been revealed that linear scaling and distribution mapping statistical bias-adjusting techniques effectively adjust the mean monthly and annual rainfall of RCMs. However, the mean-based bias-adjusting technique (linear scaling) method struggled to improve the distribution and extreme rainfall values of the RCM simulations to the observed extreme rainfall values. The distribution mapping method was effective and superior in correcting the wet-day probabilities and the 90th percentile of rainfall of RCMs to the observed rainfall. This corroborates that the distribution mapping technique has scale and shape parameters at all quantiles that can adjust the RCM simulations to observed values. However, the RCM and driving GCM types also resulted in variations in the simulation of rainfall and the models' effectiveness to reproduce the CDF, wet-day probability, and 90th percentile of the observed rainfall.

The distribution mapping method was relatively superior to the linear scaling method in preserving the climate change signal of the RCMs. This indicates higher dependence of the distribution mapping method on future RCM simulations and the bias correction functions. On the other hand, the linear scaling method strongly depended on the observed data in order to develop a scaling factor and adjust future RCM simulations. The climate change signal was also found to be dependent on the type of RCM simulations and less sensitive to emission scenarios. The REMO (MPI-ESM-LR) simulation, before and after bias adjustment, showed a positive climate change signal, preserving future RCM simulation.

The statistical bias correction techniques had different effects on extreme rainfall events. The distribution mapping method strongly affected the frequency of R1mm, R10mm, R20mm, CDD, and CWD. This could be attributed to the model-based thresholds, which adjust the frequency of wet days of climate model simulations by distribution-based bias correction methods [27]. In addition, the low frequency of R1mm days in the distribution mapping method could be due to the removal of low-intensity and drizzle rainfall events. However, the distribution mapping method showed a high frequency of R10mm

and R20mm days. This demonstrates the effect of the distribution mapping method on preserving extreme values in high quantiles. The high frequency of wet days in the linear scaling confirmed a higher number of wet days in the historical rainfall data, since the observed data highly influenced this method to generate scaling factors. There was a significant difference ($\leq 0.05$) in the number of R1mm days, R10mm, R20mm days, and CDDs according to the statistical bias correction techniques and the type of RCMs.

This study concludes that statistical bias correction methods significantly affect climate change signals and extreme rainfall values. The study has limitations, considering more RCM simulations and bias correction techniques based on spatial disaggregation. For climate change impact assessment and climate adaptation decision analysis, this study recommends using statistically bias-adjusted multi-model (E-RCMs) simulations, which show better performance in reproducing the observed climate under different robust metrics. Future research focusing on different distribution-based and spatial disaggregation bias correction techniques is essential to develop a robust climate information system for the sub-basin. The performance of bias correction techniques also needs to be evaluated using differential split sampling techniques to identify the skills of statistical bias adjusting functions under changing climate conditions.

**Supplementary Materials:** The following supporting information can be downloaded at: https://www.mdpi.com/article/10.3390/su151310513/s1, Figure S1. Boxplots (b) of mean annual rainfall of observed and RCM simulations before and after statistical bias correction in the historical period (1981–2005). The DM and LS represent distribution mapping and linear scaling bias correction methods. In the boxplots, the whiskers indicate the minimum and maximum of rainfall; the horizontal lines represent the 25th percentile, the median, and the 75th percentile from the bottom to the top of each box plot, and the diamond sign indicates outliers.

**Author Contributions:** G.W.T. was involved in the design, data analysis, discussion, and write-up of the manuscript; Y.T.D. contributed to the discussion and write-up of the manuscript; R.L.R. contributed to the discussion and write-up of the manuscript. All authors have read and agreed to the published version of the manuscript.

**Funding:** This research received no external funding.

**Institutional Review Board Statement:** Not applicable.

**Informed Consent Statement:** Not applicable.

**Data Availability Statement:** The data used in this study were accessed from the Earth System Grid Federation, which contains historical and future simulations of regional climate models. Furthermore, data can be made available upon request.

**Acknowledgments:** The authors thank the Ethiopian National Meteorological Agency for providing in situ climate data.

**Conflicts of Interest:** The authors declare that they have no known competing financial interest or personal relationship that could have appeared to influence the work reported in this manuscript.

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
