# Peer review of "Evaluating the Impact of Statistical Bias Correction on Climate Change Signal and Extreme Indices in the Jemma Sub-Basin of Blue Nile Basin"

_sustainability, doi:10.3390/su151310513_

Round 1
Reviewer 1 Report (Previous Reviewer 2)
The authors have done a good job of revising the manuscript.
Specific comments:
L58: I did not find C et al., 2010 in the references.
L203: Delete ","
L246: Check the spacing before "Besides"
Author Response
Response to Reviewer 1 Comments
We are grateful to get comments for this round of review. We believe that the comments given for this revised manuscript are important, and we have addressed the comments as described here below;
The authors have done a good job of revising the manuscript.
Specific comments:
Comments 1
L58: I did not find C et al., 2010 in the references.
Response 1
We thank the reviewer for such a thorough review. It is corrected. Please see line 58.
Comments 2
L203: Delete ","
Response 2
Corrected. Please see line 203.
Comments 3
L246: Check the spacing before "Besides"
Response 3
It is corrected. Please see line 248.
Reviewer 2 Report (Previous Reviewer 1)
This version of MS is high quality, except the following three minor issues, I have no other suggestions.
Line659-661, Please complete this reference.
Line664-666, This citation is the same as line 667-669.
Line675, Please correct 1-14 to 211.
Author Response
Response to Reviewer 2 Comments
We are thankful for this reviewer also who made a thorough review of our revised manuscript and give important comments. We have addressed the comments as described here below;
This version of MS is high quality, except the following three minor issues, I have no other suggestions.
Comments 1
Line659-661, Please complete this reference.
Response 1
We thank the reviewer for this comment. It is corrected. Please see lines 663-668.
Comments 2
Line664-666, This citation is the same as lines 667-669.
Response 2
It is corrected. Please see lines 666-668.
Comments 3
Line675, Please correct 1-14 to 211.
Response 3
We are grateful to the reviewer. It is corrected. Please see line 680.
Reviewer 3 Report (New Reviewer)
Reviewer report – Manuscript “Evaluating the impact of statistical bias correction on climate change signal and extreme indices in the Jemma sub-basin of Blue Nile Basin”
by Tefera et al.
The manuscript discusses bias correction techniques applied to regional circulation models (RCM) in Ethiopia. The topic is interesting, but I could not see elements of novelty for justifying publication. Also, the presentation of the state of the art and the structure of analysis have large margins for improvements. Finally, there are several issues on the English writing that make reading and understanding difficult at several points. Hence, my recommendation is for major revision. What follows are general and specific comments, which I sincerely hope to be useful for the authors.
General comments:
1- First and most important, a careful and thorough revision of the English writing must be performed by the authors. There are so many writing errors that it became unfeasible to point them out individually. Also, there are many confusing statements throughout the manuscript, and further attention to correct statistical terminology is required. The manuscript should be revised by a native English speaker, preferably with technical background;
2- The introductory section does not provide the necessary background for justifying the proposed technical approach. In fact, recent developments in bias correction techniques are barely addressed in the paper. The authors should clearly state the main objectives and the research gaps addressed by the study, in view of previous research. A better and more recent literature review is also necessary - most papers cited in this section date back more than 10 years and bias correction of RMCs’ outputs is a very active research topic;
3- The authors should provide proper justification for utilizing the “linear scaling” and the “distribution (quantile?) mapping” techniques. There is a plethora of models for bias correction (in a general sense, not only for RCMs), and I could see any reason for limiting the analysis to these two particular alternatives. Also, the main assumptions underlying these techniques, as well as their limitations, should be clearly presented;
4- In my opinion, the proposed approach for assessing model performance is not sufficient for supporting many of the authors’ statements. Why was only MAE utilized? Other metrics could easily disclose the limitations of the proposed bias correction approaches. This authors should elaborate this;
5- Results should always be critically discussed and not only presented. In this sense, the authors should clearly state how these results differ from previous studies – I feel that most of their findings/conclusions could be anticipated or known a priori. In addition, the actual contributions of the study to the state of the art should be clearly presented in this section. A better discussion, in the light of previous research, could also strengthen the manuscript;
6- The conclusion section does not provide a discussion per se, but merely repeat the results. What are the actual findings of the study? What are the limitations of the proposed approach? What are the envisaged research developments?
Specific comments:
1- All figure should be improved;
2- P2L33 – what do the authors mean by “non-trivial”?
3- P2L53 – “duplicate”? I do not think this term is correct;
4- P3L68 – distribution or quantile mapping?
5- P3L70 – how can bias correction techniques “reduce uncertainty”?
6- P3L85 – what do the authors mean by “develop scale and shape parameters”?
7- P3L87 – algorithms or models?
8- P4L91 – please check whether the term non-stationarity is appropriately utilized, as it refers to models or processes;
9- P4L119-120 – is it possible to “to know the specific strengths and limitations of statistical bias correction techniques” solely based on the “findings” of this study?
10- P6L144 – why do the authors refer to these particular years? Would it not be preferable to indicate the long-term streamflow?
11- P6L162 – how do the authors utilized the MICE package? It is necessary to provide details for allowing reproducibility;
12- The authors should formally define “TMAX” and “TMIN”;
13- P7L195 – “frequency and rainfall intensities” are not “rainfall values”;
14- P8L201 – Why do the authors utilized the Gamma distribution? Shouldn’t a goodness-of-fit assessment be performed for model selection?
15- P8L228 – box plots are exploratory tools, which are not designed for model assessment;
16- P9L238 – the authors should explain why they chose this period;
17- P9L255-256 – mm/day?
18- P11L296-297 – how were the differences in MAE evaluated (which test)?
19- P14L344-345 and P15L346 – these statements are too confusing. Please clarify;
20- P17L387-388 – wasn’t that known a priori?
21- P18L404 – what do the authors mean by “high scaling parameters”?
22- P18L410-412 – these statements are too confusing. Please clarify;
23- P18L425-426 – wasn’t that known a priori?
24- P18L427 – how were the differences evaluated (which test)?
25- P18L429-431 – these statements are too confusing. Please clarify;
26- P21L477-478 – how do the “distribution mapping method applies scale parameters that remove low-intensity rainfall days”?
27- P22L496-497 – what do the authors mean by “better scale and shape parameters”?
28- P22L522-523 – wasn’t that known a priori?
29- P22L525-526 – how can the authors recommend these methods when they were not evaluated? I am a bit confused.

Extensive editing of the English writing is necessary.
Author Response
Responses to reviewer
We appreciate the thoughtful comments given by the reviewer. We believe that the comments given are important and our manuscript has benefited a lot from this round of revision. We have addressed the comments as described here below;
General comments:
Comments 1
First and most important, a careful and thorough revision of the English writing must be performed by the authors. There are so many writing errors that it became unfeasible to point them out individually. Also, there are many confusing statements throughout the manuscript, and further attention to correct statistical terminology is required. The manuscript should be revised by a native English speaker, preferably with technical background;
Response 1
We thank the reviewer for this important comment. We made a thorough revision on the manuscript with our language editor. This comment helps us to see each sentence of the manuscript. We also thank our language editor.
Comments 2
The introductory section does not provide the necessary background for justifying the proposed technical approach. In fact, recent developments in bias correction techniques are barely addressed in the paper. The authors should clearly state the main objectives and the research gaps addressed by the study, in view of previous research. A better and more recent literature review is also necessary - most papers cited in this section date back more than 10 years and bias correction of RMCs’ outputs is a very active research topic;
Response 2
We are grateful for this comment which helps us to incorporate recent articles on statistical bias correction. We made a thorough revision to the introduction section. Please see the track change on the introduction section. The following are among the articles we have added in the introduction and other sections of the manuscript;
Zhang, J., Peng, S., Wang, Z., Fu, J., Li, Z., 2023. Daily precipitation and temperature for 2021–2050 over China: Multiple RCMs and emission scenarios corrected by a trend-preserving method. Int. J. Climatol. 1955–1969. https://doi.org/10.1002/joc.7955
Zhu, L., Kang, W., Li, W., Luo, J.J., Zhu, Y., 2022. The optimal bias correction for daily extreme precipitation indices over the Yangtze-Huaihe River Basin, insight from BCC-CSM1.1-m. Atmos. Res. 271, 106101. https://doi.org/10.1016/j.atmosres.2022.106101
Chen, J., Yang, Y., Tang, J., 2022. Bias correction of surface air temperature and precipitation in CORDEX East Asia simulation: What should we do when applying bias correction? Atmos. Res. 280, 106439. https://doi.org/10.1016/j.atmosres.2022.106439
Tegegne, G., Melesse, A.M., 2021. Comparison of Trend Preserving Statistical Downscaling Algorithms Toward an Improved Precipitation Extremes Projection in the Headwaters of Blue Nile River in Ethiopia. Environ. Process. 8, 59–75. https://doi.org/10.1007/s40710-020-00474-z
IPCC, 2021. Climate Change 2021: The Physical Science Basis. Contribution of Working Group I to the Sixth Assessment Report of the Intergovernmental Panel on Climate Change [Masson-Delmotte, [Masson-Delmotte, V., P. Zhai, A. Pirani, S.L. Connors, C. Péan, S. Berger, , Cambridge University Press, Cambridge, United Kingdom and New York, NY, USA,.
Su, T., Chen, J., Cannon, A.J., Xie, P., Guo, Q., 2020. Multi-site bias correction of climate model outputs for hydro-meteorological impact studies: An application over a watershed in China. Hydrol. Process. 34, 2575–2598. https://doi.org/10.1002/hyp.13750
Casanueva, A., Herrera, S., Iturbide, M., Lange, S., Jury, M., Dosio, A., Maraun, D., Gutiérrez, J.M., 2020. Testing bias adjustment methods for regional climate change applications under observational uncertainty and resolution mismatch. Atmos. Sci. Lett. 21, 1–12. https://doi.org/10.1002/asl.978
Gnitou, G.T., Tan, G., Ma, T., Akinola, E.O., Nooni, I.K., Babaousmail, H., Al-Nabhan, N., 2021. Added value in dynamically downscaling seasonal mean temperature simulations over West Africa. Atmos. Res. 260, 105694. https://doi.org/10.1016/j.atmosres.2021.105694
Eum, H. Il, Cannon, A.J., 2017. Intercomparison of projected changes in climate extremes for South Korea: application of trend preserving statistical downscaling methods to the CMIP5 ensemble. Int. J. Climatol. 37, 3381–3397. https://doi.org/10.1002/joc.4924
Cannon, A.J., Sobie, S.R., Murdock, T.Q., 2015. Bias correction of GCM precipitation by quantile mapping: How well do methods preserve changes in quantiles and extremes? J. Clim. 28, 6938–6959. https://doi.org/10.1175/JCLI-D-14-00754.1
Rummukainen, M., 2016. Added value in regional climate modeling. Wiley Interdiscip. Rev. Clim. Chang. 7, 145–159. https://doi.org/10.1002/wcc.378
Comments 3
The authors should provide proper justification for utilizing the “linear scaling” and the “distribution (quantile?) mapping” techniques. There is a plethora of models for bias correction (in a general sense, not only for RCMs), and I could see any reason for limiting the analysis to these two particular alternatives. Also, the main assumptions underlying these techniques, as well as their limitations, should be clearly presented;
Response 3
We thank the reviewer for this important comment. Yes, there are plethora of statistical bias correction techniques. But, we consider the linear scaling technique because it is a mean-based statistical bias method and the distribution mapping method because it adjusts the cumulative distribution function of climate model simulations towards the cumulative distribution function of observed values. The characteristics of the methods considered in this study are quite different and represent various mean-based and distribution-based bias correction techniques. In our future work, we will consider different new statistical bias correction techniques and statistical packages.
Comments 4
In my opinion, the proposed approach for assessing model performance is not sufficient for supporting many of the authors’ statements. Why was only MAE utilized? Other metrics could easily disclose the limitations of the proposed bias correction approaches. This authors should elaborate this;
Response 4
We are grateful for the comment. The manuscript has used other metrics to evaluate regional climate models before and after bias correction. In addition to Mean Absolute Error (MAE), we have used Cumulative Distribution Function (CDF), probability of wet days (Prwet (%)), and 90th percentile (X90 (mm)). Please see section 2.6.
Comments 5
Results should always be critically discussed and not only presented. In this sense, the authors should clearly state how these results differ from previous studies – I feel that most of their findings/conclusions could be anticipated or known a priori. In addition, the actual contributions of the study to the state of the art should be clearly presented in this section. A better discussion, in the light of previous research, could also strengthen the manuscript;
Response 5
We thank the reviewer for this comment. We made exhaustive revisions to the result section of the manuscript. Please see the revision on the track version of the manuscript.
Comments 6
The conclusion section does not provide a discussion per se, but merely repeat the results. What are the actual findings of the study? What are the limitations of the proposed approach? What are the envisaged research developments?
Response 6
We found it is an important comment. Accordingly, we have revised the conclusion section of the manuscript. We have incorporated the main findings, the limitations and recommendations on future research areas. Please see the track changes in the conclusion section.
Specific comments:
Comments 1
All figure should be improved;
Response 1
We thank the reviewer for such a thorough review. We have modified most of the figures. Please see figures in the revised version of the manuscript. However, we have seen the quality of the figures in the PDF version of our submission is distorted. We suggest the reviewer to see the Word version of our submission.
Comments 2
P2L33 – what do the authors mean by “non-trivial”?
Response 2
To say “important” or “essential”. It is corrected.
Comments 3
P2L53 – “duplicate”? I do not think this term is correct;
Response 3
It is corrected.
Comments 4
P3L68 – distribution or quantile mapping?
Response 4
distribution mapping
Comments 5
P3L70 – how can bias correction techniques “reduce uncertainty”?
Response 5
Bias correction methods reduce uncertainty by adjusting systematic biases in the RCM simulations.
Comments 6
P3L85 – what do the authors mean by “develop scale and shape parameters”?
Response 6
Statistical bias correction techniques develop scale and shape parameters. Using the raw climate model simulations and observed values, bias correction techniques develop scale parameters that adjust the magnitude of RCM simulations and shape parameters which adjust the distribution of values of RCM simulations. This means the scale parameters adjust over and underestimation by RCM simulations while shape parameters adjust frequency of different values such as frequency of high and low values simulated by RCMs.
Comments 7
P3L87 – algorithms or models?
Response 7
Algorithms (functions) that are used to adjust RCM simulations.
Comments 8
P4L91 – please check whether the term non-stationarity is appropriately utilized, as it refers to models or processes;
Response 8
We thank the reviewer for such a thorough review. Yes, non-stationarity in this sentence refers to the climate model simulations.
Comments 9
P4L119-120 – is it possible to “to know the specific strengths and limitations of statistical bias correction techniques” solely based on the “findings” of this study?
Response 9
We agree it is hardly possible to know all strengths and limitations of bias correction techniques. In this manuscript, we evaluate the specific strengths and limitations. The manuscript has evaluated; 1) the skill of statistical bias correction methods in adjusting the historical climate with observed data, 2) how statistical bias correction techniques are effective in adjusting extreme values and frequencies in the historical and future period and 3) how statistical bias correction techniques are effective in preserving/changing the extreme values and frequency distribution in RCM simulations.
Comments 10
P6L144 – why do the authors refer to these particular years? Would it not be preferable to indicate the long-term streamflow?
Response 10
We thank the reviewer for this comment. We mentioned these years because measured streamflow data (better quality streamflow data) is available only for these years.
Comments 11
P6L162 – how do the authors utilized the MICE package? It is necessary to provide details for allowing reproducibility;
Response 11
We thank the reviewer for this comment. We have added a description of how we have used the MICE package. Please see line 255-259 on the track version of the manuscript.
Comments 12
The authors should formally define “TMAX” and “TMIN”;
Response 12
It is corrected. Please see line 262 on the track version of the manuscript.
Comments 13
P7L195 – “frequency and rainfall intensities” are not “rainfall values”;
Response 13
It is rewritten. Please see line 300.
Comments 14
P8L201 – Why do the authors utilized the Gamma distribution? Shouldn’t a goodness-of-fit assessment be performed for model selection?
Response 14
The gamma distribution is effective and the recommended goodness-of-fit assessment for precipitation (Teutschbein and Jan Seibert, 2012). Thus, we have used the gamma distribution to evaluate precipitation distribution before and after bias correction.
Comments 15
P8L228 – box plots are exploratory tools, which are not designed for model assessment;
Response 15
We thank the reviewer for this comment. Even though we have used other statistical metrics such as Cumulative Distribution Function (CDF), Mean Absolute Error (MAE), probability of wet days (Prwet (%)), and 90th percentile (X90 (mm)), box plots is commonly used method in this research area. A box plot is common to show outlier, minimum, maximum, percentile, and median values. Please see similar studies, for instance, Cannon et al., 2015; Wootten et al., 2021; Holthuijzen et al., 2022; Hernanz et al., 2021.
Comments 16
P9L238 – the authors should explain why they chose this period;
Response 16
The historical period (1981-2005) is chosen because of 1) the historical RCM simulations are until 2005, 2) better quality observed data was available from 1981-2014. The period 2071-2100 is selected because this time frame is commonly used in climate change projections (Cannon et al., 2015; Mbaye et al., 2016; Wootten et al., 2021).
Comments 17
P9L255-256 – mm/day?
Response 17
Yes
Comments 18
P11L296-297 – how were the differences in MAE evaluated (which test)?
Response 18
The paired t-test was used to determine whether there is a significant difference between the MAE values of observed and RCM simulations before and after bias correction.
Comments 19
P14L344-345 and P15L346 – these statements are too confusing. Please clarify;
Response 19
Thank you, this has been rewritten.
Comments 20
P17L387-388 – wasn’t that known a priori?
Response 20
The main intent of the sentence is to indicate other sources of uncertatinity.
Comments 21
P18L404 – what do the authors mean by “high scaling parameters”?
Response 21
This is to indicate the factor used to adjust the RCM simulations. For instance, distribution mapping method may use scaling parameter value of 0.7, 1 or 1.5. The parameter value 0.7 is used when the RCM simulation show overestimation and needs to be adjusted by the scaling parameter value of 0.7 to fit with the observed value. The parameter value 1 is used when the RCM simulation is equal to the observed value and needs no modification. The parameter value 1.5 is used when the RCM simulation show underestimation and needs to be adjusted by the scaling parameter value of 1.5 to fit with the observed value.
Comments 22
P18L410-412 – these statements are too confusing. Please clarify;
Response 22
Thank you, this has been rewritten. Please see line 686-688.
Comments 23
P18L425-426 – wasn’t that known a priori?
Response 23
Comments 24
P18L427 – how were the differences evaluated (which test)?
Response 24
Using paired t-test, whether the difference was significant or not signanificant (at p≤0.05) was evaluated.
Comments 25
P18L429-431 – these statements are too confusing. Please clarify;
Response 25
Thank you, this has been rewritten. Please see line 718-720.
Comments 26
P21L477-478 – how do the “distribution mapping method applies scale parameters that remove low-intensity rainfall days”?
Response 26
We thank the reviewer for this comment. One of the advantage of distribution mapping method over linear scaling method is it develop precipitation threshold to adjust the frequency of wet-days. In this study, there were more number of days with drizzle rainfall (<1mm/day) in the RCM simulations. Thus, the precipitation threshold does not consider these drizzle precipitation days as wet days.
Comments 27
P22L496-497 – what do the authors mean by “better scale and shape parameters”?
Response 27
This means the parameters (factors) used to adjust the intensity (scale) and distribution (shape) of RCM simulatios toward the observed counter parts are effective. In this case the mean annual and monthly values, extreme values and the frequency of low and high rainfall values of bias corrected and observed
Comments 28
P22L522-523 – wasn’t that known a priori?
Response 28
What is largely known is statistical bias correction methods adjust climate model simulations particularly mean values in the historical period. However, the effect of bias correction on climate change signal and extreme values less explored and the existing literature are not conclusive in this regard.
Comments 29
P22L525-526 – how can the authors recommend these methods when they were not evaluated? I am a bit confused.
Response 29
We are grateful for this comment. If we did not evaluate the skill of these bias correction techniques on climate change signal and extreme values, our recommendation was to be only based on the performance of the bias correction techniques in the historical climate. It was not possible to recommend distribution mapping method in preserving climate change signal and extreme values. in this study. We have seen the change on climate change signal and extreme indices is also attributed by the RCM types, which was hardly possible to know before this study.

Round 2
Reviewer 3 Report (New Reviewer)
The manuscript has considerably improved with respect to the original submission. However, some points still must be addressed by the authors. First, there are still many writing errors and incorrect terminology. For instance, statements like “high scaling parameters” and “better scale and shape parameters” do not make sense. The authors should clarify these and other points or utilize correct terminology. Also, the authors should provide stronger justification for the adopted bias-correction techniques along the text, not only in their responses to review reports. Finally, I am not convinced that MAE and the 90th percentile are sufficient for assessing bias correction approaches with respect to extreme values. I think more extreme percentiles and other metrics, such as RMSE, are useful in this context and would strengthen the analysis of the results.
Moderate editing is required.
Author Response
Responses to reviewer
We have great appreciation for the reviewer who gives important comments for this round of revision. We believe that the comments given are important and our manuscript has benefited a lot from this round of revision. We have addressed the comments as described below.
Comments 1
The manuscript has considerably improved with respect to the original submission. However, some points still must be addressed by the authors. First, there are still many writing errors and incorrect terminology. For instance, statements like “high scaling parameters” and “better scale and shape parameters” do not make sense. The authors should clarify these and other points or utilize correct terminology
Response 1
We thank the reviewer for such a thorough review. We have corrected it. Please see lines 615 and 745.
Comments 2
The authors should provide stronger justification for the adopted bias-correction techniques along the text, not only in their responses to review reports.
Response 2
We are grateful for this comment. We have incorporated this comment. Please see lines 295-304.
Comment 3
Finally, I am not convinced that MAE and the 90th percentile are sufficient for assessing bias correction approaches with respect to extreme values. I think more extreme percentiles and other metrics, such as RMSE, are useful in this context and would strengthen the analysis of the results.
Response 3
We thank the reviewer for this comment. This study has used different mean and distribution-based metrics to analyze the skill of the statistical bias correction technique and the change in climate signal and extreme indices due to statistical bias correction techniques. For this, we have used mean, MAE, 90th percentile, Cumulative Distribution Function (CDF), wet day probability, number of heavy precipitation days (R10mm), number of very heavy rainfall days (R20mm), Consecutive Dry Days (CDD) and Consecutive Wet Days (CWD). We believe the statistical metrics used to study extreme precipitation are adequate.
Comment 4
Careful and thorough revision of the English writing must be performed by the authors. There are so many writing errors that it became unfeasible to point them out individually. Also, there are many confusing statements throughout the manuscript, and further attention to correct statistical terminology is required. The manuscript should be revised by a native English speaker, preferably with technical background;
Response 4
We are grateful for this comment. We made a thorough revision on this version of the manuscript. We have corrected some grammatical and structure of the language.

This manuscript is a resubmission of an earlier submission. The following is a list of the peer review reports and author responses from that submission.
Round 1
Reviewer 1 Report
The paper Tefera et al. analyzed the statistical bias correction of regional climate model simulation for cliamte change in Jemma sub-basin. This MS is interesting and under the scope of the journal 'SUSTAINABILITY'. The article is well constructed, the experiments were well conducted, and analysis was well performed. However, I still have the following quesitons in this study:
(1) The study fails to address the scientific questions and hypothesis clearly.
(2) What is the significance of the research in this study, and what is the theoretical value and application value of the research? It is recommended that the author briefly explain in the abstract.
(3) The reference is too old, the author needs to update the recent literature.
Reviewer 2 Report
The authors have studied the impact of statistical bias correction on climate change signal and extreme indices. This is an important aspect of the changing climate. Overall, it is good written manuscript. However, I wonder authors did not cover the latest research in this direction. I found two interesting papers on bias correction and the impact of climate change due to data resolution. I recommend amending the below publications and discussing the comparison between these studies with yours.
Temporal disaggregation of hourly precipitation under changing climate over the Southeast United States (https://www.nature.com/articles/s41597-022-01304-7)
Projected mid-century rainfall erosivity under climate change over the southeastern United States (https://www.sciencedirect.com/science/article/pii/S0048969722082225)
Check the citation style throughout the manuscript, such as L98, 102, and throughout the manuscript. you are missing commas in most of your citations.
Specific comments are provided below:
L11: Add a sentence for the method before explaining the results.
L61-65: Add current citations and references. Check what I have listed.
L93-95: Re-write the sentence. Possibly, split the sentence for smooth reading.
L112: Add “to” before “examine”.
Figure 1: check coordinates and make them visible. Check legend Elevation: is it -161?
L146: 3000 m.a.s.l
L172: what is RCA4? Is it RCM4? If it is RCM, the authors have already defined it as an abbreviation.
L250- Similar results of overestimation of bias-corrected mean annual rainfall were reported by Takhellambam et al. (2022, 2023). I strongly recommend these references to include in your study. I find them very interesting since the work is very significant in your manuscript.
Reviewer 3 Report
Utilizing two bias correction techniques, this manuscript compared the their effects on several climate change signals and extreme rainfall indices in historical period, and in future period under two emission scenarios. Relative performances of these two bias correction methods were evaluated and their significant effects were confirmed.
The authors have made endeavors in preparing the manuscript,which was appreciated by the reviewer. But major concerns should be addressed:
First of all, the manuscript is more like a technical report, which might give instructions on selection of bias correction method while failed to show importance or value in science. Even, the evaluation work in the research area(Jemma sub-basin) maynot be representative in other parts of the country or world. It is very important for the authors to sort out the question, like, why is this work worthy of noticing or reading by other people,and what is the contribution of this work to science research.
Secondly, the evaluation methods are insufficient in the manuscript, barely comparing the outputs of the two bias correction methods with observed data. The box plot is not an evaluation method actually.
Thirdly, there are several improper expressions in writing and figures in the manuscript, making it important for the authors to get systemetic training in writing of scientific research papers.
Specific comments
Line 109:the punctuation mark ‘;’->’:’
Line 130: there should not be brackets for 9 ℃
Line 150:re-write the sentence ‘Nine climatic station ……’, it’s wrong in grammer.
Line 181: the punctuation mark ‘;’->’:’
Line 182\201: the expression of two equations are not unified;the number of equations should be in bracket or referenced as Eq 1
Sect.2.6(line 219-237): This section is not necessary, there are no specific evaluation methods, and these comparison methods have been mentioned in Sect.3.
Figure2:each sub-figure should be given a specific label, like (a-1)\(a-2) etc., or like the label of Figure 6.
Line265-268:these explanations to the Figure2 are recommened to be in the caption of Figure 2. ‘the circles’ in line 266 should be ‘the whiskers’
Figure3:each sub-figure should be given a specific label, like (a)\(b) etc..
Line 378\392\395: ‘ R10m’ should be ‘R10mm’ according to the authors logic,please check the correct all the wrong usages.